Shot-gun proteome and transcriptome mapping of the jujube floral organ and identification of a pollen-specific S-locus F-box gene

Chen Ruihong 1
Chen Guoliang 1
Huang Jian huangj@nwsuaf.edu.cn 2
1 Shaanxi Province Key Laboratory of Jujube, College of Life Science, Yan’an University , Yan’an , China
2 College of Forestry, Northwest A & F University , Yangling , China
Liang Chun
Electronic publication date: 2017 Jul 17
Publication date: 2017
Volume: 5
Electronic Location ID: e3588
Received 2017 Jan 23; Accepted 2017 Jun 27
Copyright: ©2017 Chen et al.
Copyright year: 2017
Copyright holder: Chen et al.
License: This is an open access article distributed under the terms of the Creative Commons Attribution License, which permits unrestricted use, distribution, reproduction and adaptation in any medium and for any purpose provided that it is properly attributed. For attribution, the original author(s), title, publication source (PeerJ) and either DOI or URL of the article must be cited.
License URL: https://creativecommons.org/licenses/by/4.0/

Keywords: Ziziphus jujuba, Floral biology, Transcriptome, Proteome, Gametophytic self-incompatibility, S-locus F-box gene

Funding: Shanxi Provincial Education Department 16JS117 This study was supported by doctoral research funds from Yan’an University, the Key Laboratory project fund of Shanxi Provincial Education Department (Grant No. 16JS117), and doctoral research funds from Shaanxi Province to Dr. Huang. The funders had no role in study design, data collection and analysis, decision to publish, or preparation of the manuscript.

==============================
The flower is a plant reproductive organ that forms part of the fruit produced as the flowering season ends. While the number and identity of proteins expressed in a jujube (Ziziphus jujuba Mill.) flower is currently unknown, integrative proteomic and transcriptomic analyses provide a systematic strategy of characterizing the floral biology of plants. We conducted a shotgun proteomic analysis on jujube flowers by using a filter-aided sample preparation tryptic digestion, followed by liquid chromatography-tandem mass spectrometry (LC-MS/MS). In addition, transcriptomics analyses were performed on HiSeq2000 sequencers. In total, 7,853 proteins were identified accounting for nearly 30% of the ‘Junzao’ gene models (27,443). Genes identified in proteome generally showed higher RPKM (reads per kilobase per million mapped reads) values than undetected genes. Gene ontology categories showed that ribosomes and intracellular organelles were the most dominant classes and accounted for 17.0% and 14.0% of the proteome mass, respectively. The top-ranking proteins with iBAQ >1010 included non-specific lipid transfer proteins, histones, actin-related proteins, fructose-bisphosphate aldolase, Bet v I type allergens, etc. In addition, we identified one pollen-specificity S-locus F-box-like gene located on the same chromosome as the S-RNase gene. Both of these may activate the behaviour of gametophyte self-incompatibility in jujube. These results reflected the protein profile features of jujube flowers and contributes new information important to the jujube breeding system.

Introduction

Chinese jujube (Ziziphus jujuba Mill.) is one of the most popular fruit trees in China. The fruits are valued for their good taste and their high nutritional and medical value (Liu et al., 2014; Huang et al., 2016). The increasing economic value of the Chinese jujube has resulted in an increase in cultivation acreage.

Flowers are a critical reproductive organ and are the origin of these valuable fruits. They begin to develop in early June, originating from the leaf axil with shoot growth, and continue to be produced until the end of October in northern China (Niu, Zhang & Yuan, 2015). Jujube flowers also produce abundant nectar, and honey made from jujube nectar is one of the most popular types of honey in China. However, jujube flowers are small and drop easily; 87.9–99.9% of the flowers and 68.5% of young fruits usually drop during development, and only 1.1% of the fruits reach maturity on average (Liu et al., 2009). Therefore, understanding jujube floral biology is important and may provide insights into fruit setting and jujube breeding. Although several studies have focused on the biology of flower differentiation (He, Zhang & Jiang, 2009; Song et al., 2013; Niu, Peng & Li, 2011), a global analysis of protein expression in jujube flowers is needed to understand floral biology.

Proteomics provide a comprehensive and quantitative approach to identify proteins that are expressed in a given organ, tissue, or cell line. With the advance in high-resolution mass spectrometry (MS)—based shotgun proteomics (Mann, Kulak & Nagaraj, 2013), several complete proteome maps have been assembled in recent years. However, these studies were restricted to model organisms (Wilhelm et al., 2014; Kim et al., 2014; Baerenfaller et al., 2008). Recently, whole-genome sequencing has been applied to more organisms, and the developed gene models have provided information on the theoretical protein-coding capacity of an organism. These advances have enabled complete proteome analysis for more species. Since the first proteome map of a model plant species, Arabidopsis thaliana, was assembled (Baerenfaller et al., 2008), rapid progress in plant proteome has been made in various plant species at different tissue levels, such as fruit, seeds, flowers and leaves roots (Zeng et al., 2011; Girolamo, D’Amato & Righetti, 2012; Dai et al., 2013; Zhu et al., 2015). Undoubtedly, sequencing of the jujube genome will also contribute to the proteomics study of jujube (Liu et al., 2014).

Some genes specifically expressed in floral organs have been shown to contribute to flower development and reproductive function. In a previous study, gametophyte self-incompatibility (GSI) induced in Z. jujuba led to high percentage of seedless fruits (Asatryan & Tel-Zur, 2013). Self-incompatibility/compatibility is controlled by the pistil S determinant ribonuclease (S-RNase) and the pollen S determinant F-box (SFB) gene(s). The GSI has been well characterized in some core-eudicots, such as Solanaceae, Plantaginaceae and Rosaceae. We have previously identified an S-RNase candidate gene in the chromosome 1 of the assembled ‘Junzao’ jujube genome (Huang et al., 2016). However, the SFB candidate genes specifically expressed in pollens have not been identified.

A combination of transcriptome and proteome technologies offers a new comprehensive approach to identify the genes involved in developmental systems and environmental responses. This study is the first to generate a draft proteome map of jujube flowers based on the liquid chromatography-tandem mass spectrometry (LC-MS/MS). To further improve the molecular knowledge of GSI in Z. jujuba, we attempt to identify the SFB gene(s) specially expressed in anthers/pollens linked to the S-RNase gene. Information from our study will provide a better understanding of jujube flower development and the breeding system.

Materials and Methods

Sample collection

Budding flowers were collected from nine year old jujube trees (Ziziphus jujuba ‘Junzao’) in 2013. We collected two biological replicates of flower samples from each of two trees. The collected flowers were immediately immersed in liquid nitrogen, and stored at −80 °C for transcriptome and proteome analysis.

Protein extraction and quality determination

We selected one sample replicate for proteome analysis. The trichloroacetic acid (TCA)/acetone precipitation method was used to extract protein from jujube flowers (Wu et al., 2014). Briefly, frozen flowers were ground in liquid nitrogen and precipitated with TCA/acetone (1:9) overnight at −20 °C. Then, the homogenate was centrifuged (7,000 × g, 30 min) and washed three times with acetone. The precipitated proteins were freeze-dried and then re-suspended in 800 µl of SDT buffer (4% SDS, 100 mM DTT, 150 mM Tris–HCl, pH 8.0). After 15 min of incubation in boiling water, the homogenate was fractionated by ultrasound (80 W, 10 s, 10 times) on ice. The homogenate was centrifuged at 14,000 × g at 4 °C for 30 min, and the supernatant was collected. Finally, the protein concentration was determined using a bicinchoninic acid protein assay kit (BCA, Beyotime, China) and adjusted to 5 µg/µl. Protein was also analysed by electrophoresis on a 12.5% sodium dodecyl sulphate polyacrylamide gel electrophoresis (SDS-PAGE) gel (7 cm  × 20 cm) and stained with Coomassie Brilliant Blue (CBB) staining buffer (GelCode blue; Pierce, Waltham, MA, USA).

In-solution protein digestion

Proteins were digested by following the filter-aided sample preparation (FASP) procedure (Wiśniewski, Zougman & Nagaraj, 2009). Total-protein samples (50 µg) were incorporated into 30 µl of SDT buffer (4% SDS, 100 mM DTT, 150 mM Tris–HCl, pH 8.0) and then heated at 95 °C for 5 min. Thereafter, 200 µl of UA buffer (8 M urea, 150 mM Tris–HCl pH 8.0) was added to the protein solution, mixed to remove the detergent, DTT, and passed through an ultrafiltration unit with a nominal molecular weight (MW) cut-off of 30 kDa (Cat No. MRCF0R030, Millipore) to remove other low-molecular-weight components. Then, 100 µl of 0.05 M iodoacetamide in UA buffer was added, and the samples were incubated for 20 min in the dark. The filter was washed three times with 100 µl of UA buffer, followed by two washes with 100 µl of 25 mM NH4HCO3. Finally, 2 µg of trypsin (Promega, Madison, WI, USA) was added, the protein suspension was digested overnight at 37 °C and the resulting peptides were collected as the filtrate.

Peptide fractionation with strong cation exchange (SCX) chromatography

Peptides were fractionated by SCX chromatography using an AKTA Purifier system (GE Healthcare). The dried peptide mixture was reconstituted and acidified with 2 ml of buffer A (10 mM KH2PO4 in 25% of acetonitrile (ACN), pH 2.7) and loaded onto a Poly SULFOETHYL 4.6 × 100 mm column (5 µm, 200 Å; PolyLC Inc, Maryland, USA). The peptides were eluted at a flow rate of 1 ml/min using a gradient of 0–10% buffer B (500 mM KCl, 10 mM KH2PO4 in 25% of ACN, pH 2.7) for 2 min, 10–20% buffer B for 25 min, 20–45% buffer B for 5 min, and 50%–100% buffer B for 5 min. The eluents were monitored by absorbance at 214 nm and fractions were collected every 1 min. The collected fractions (approximately 30) were combined into 25 pools and desalted on C18 Cartridges (Empore™ SPE Cartridges C18 (standard density), bed I.D. 7 mm, volume 3 ml; Sigma Aldrich, St. Louis, MO, USA). Each fraction was concentrated by vacuum centrifugation and reconstituted in 40 µl of 0.1% (v/v) trifluoroacetic acid. All samples were stored at −80 °C until LC-MS/MS analysis could be conducted.

Liquid Chromatography (LC)—Electrospray ionization (ESI) tandem MS (MS/MS) analysis by Q Exactive

Experiments were performed on a Q Exactive mass spectrometer that was coupled to a nanoflow HPLC (Easy nLC; Proxeon Biosystems/Thermo Fisher Scientific, Waltham, MA, USA). Each fraction (6 µl) was injected for Nano LC-MS/MS analysis. The peptide mixture (5 µg) was loaded onto a C18-reversed phase column (Thermo Scientific Easy Column, 10 cm long, 75 µm inner diameter, 3 µm resin) in buffer A (0.1% formic acid) and separated using a linear gradient of buffer B (80% acetonitrile and 0.1% formic acid) at a flow rate of 250 nl/min controlled by IntelliFlow technology over 60 min.

MS data were acquired dynamically using a data-dependent top10 method by choosing the most abundant precursor ions from the survey scan (300–1,800 m/z) for HCD fragmentation. Target value determination was based on predictive automatic gain control (pAGC). The dynamic exclusion duration was 60 s. Survey scans were acquired at a resolution of 70,000 at 200 m/z , and for HCD spectra, the resolution was set to 17,500 at 200 m/z. The normalized collision energy was 30 eV, and the underfill ratio, which specifies the minimum percentage of the target value likely to be reached at maximum fill time, was defined as 0.1%. The instrument was run in peptide recognition mode.

RNA extraction and Transcriptome analyses

Total RNA of whole budding flowers was isolated using an RNA prep Pure Tissue Kit (Tiangen, Beijing, China) for RNA extraction and then treated with RNase-free DNase I treatment (Promega, Madison, WI, USA). RNA purity was checked using the NanoPhotometer® spectrophotometer (Implen, Westlake Village, CA, USA). RNA concentration was measured using Qubit 2.0 Flurometer (Life Technologies, Carlsbad, CA, USA). RNA integrity was assessed using an RNA Nano 6000 Assay Kit of the Bioanalyzer 2100 system (Agilent Technologies, Santa Clara, CA, USA). Finally, RNA degradation and contamination was visualized on 1% agarose gels. A total of 3 µg of RNA per sample was used as input material for the RNA sample preparations. Sequencing libraries were generated using NEBNext® Ultra™ RNA Library Prep Kit for Illumina® (NEB, Ipswich, MA, USA) according to manufacturer’s recommendation. The libraries were sequenced on an Illumina HiSeq 2000 platform and 100 bp paired-end reads were generated. The raw-sequence reads data are deposited in NCBI Sequence Read Archive (SRX1518648 and SRX1518646).

After quality control of raw reads, filtered reads were mapped to the ‘Junzao’ genome using Tophat2 (Kim et al., 2013). Total aligned read numbers were normalized by gene length and sequencing depth for an accurate estimate of expression levels (Robinson & Oshlack, 2010). HTSeq 0.6.1 was used to count the reads mapping to each gene (Anders, Pyl & Huber, 2014). In addition, the RPKM (reads per kilobase per million mapped reads) of each gene was calculated based on the length of the gene to represent the expression level (Mortazavi et al., 2008).

Sequence database searching and data analysis

The mass spectrometric raw data from the top10 method were analysed using MaxQuant 1.3.0.5 (Cox & Mann, 2008). MS/MS data were searched against the gene predicted gene models of Ziziphus jujuba ‘Junzao’ (Accession: LPXJ00000000 and Dryad Digital Repository. doi: 10.5061/dryad.83fr7). An initial search was set at a precursor mass window of 6 ppm. The search followed an enzymatic cleavage rule of Trypsin/P and allowed a maximum of two missed cleavage sites and a mass tolerance of 20 ppm for fragment ions. Enzyme = Trypsin, Missed cleavage = 2, Fixed modification: Carbamidomethyl (C), Variable modification: Oxidation (M), Decoy database pattern = decoy. The cut-off of global false discovery rate (FDR) for peptide and protein identification was set to 0.01. The iBAQ algorithm was used to calculate the approximate abundance of each protein, which normalizes the summed peptide intensities by the number of theoretically observable peptides of the protein (Schwanhäusser et al., 2013).

Gene Ontology (GO) term enrichment analysis of expressed genes and proteins

GO term enrichment analysis was conducted for expressed genes and proteins in the jujube flower using the R package TopGO, and is available from Dryad Digital Repository (doi: 10.5061/dryad.83fr7). GO term enriched results for the transcriptome and proteome were compared for the three aspects of cell components, molecular function and biological progress.

Gene expression validation by quantitative Real-time PCR (qRT-PCR) analysis

We performed a qRT-PCR analysis to validate the relative gene expression levels in the flowers. RNA samples from the same flower samples used for RNA-sequencing analysis were extracted using the same method as described above. cDNA was synthesized using a PrimeScript™ RT reagent kit with gDNA Eraser (TaKaRa, Dalian, China) according to the manufacturer’s instruction. We selected 30 genes with different levels of RPKM and iBAQ values for qRT-PCR analyses (Table S1). The primers were designed using Primer-BLAST online software (http://www.ncbi.nlm.nih.gov/tools/primer-blast/). Primer specificities were initially verified by PCR amplification of a single product and product size was visualized after electrophoresis on a 1.2% agarose gel. The cDNAs were then amplified in a Bio-Rad CFX Real-Time System (Bio-Rad, Hercules, CA, USA) with SYBR Premix Ex Taq TM II (Takara, Dalian, China). The qRT-PCR amplification protocol was set at 95 °C for 3 min, 95 °C for 10 s, 40 cycles of 95 °C for 10 s and 58 °C for 10 s, and finally, 72 °C for 30 s. Three replicates were performed for each gene. The UBQ gene (Zj.jz021445712) of jujube was chosen as an internal reference gene for normalization (Zhang, Huang & Li, 2015). The average threshold cycle (Ct) was normalized, and the relative expression was calculated using the 2−ΔCt method.

Identification of S-locus F-box (SFB) genes

We used previously reported S-locus F-box genes as queries to BLAST search ‘Junzao’ gene models, such as those from Plantaginaceae (6 S-locus F-box (SLF) genes), Solanaceae (35 SLFs), and Maleae (22 S-locus F-box brothers (SFBB)), 8 SLF-like genes and 3 SFB genes of Prunus. Because we have identified the S-RNase gene (Zj.jz035833030) located on the linkage group (LG) 1 of the assembled ‘Junzao’ genome, we treated the newly identified F-box genes located on LG1 as potential SFB genes and then subjected them to phylogenetic analysis with the known reported S-locus F-box genes. All the amino acid sequences were aligned and a neighbour-joining tree was constructed using Mega 6.0.6 (Tamura et al., 2013). Then, RPKM values of those F-box genes were calculated based on the transcriptomic analysis of leaves, flowers, shoot and fruits based on the previous report (Huang et al., 2016). We further performed tissue-specific expression analysis of those F-box genes among different tissues/organs (leaves, shoot, young fruit, ripe fruit, pollen, pistil, petal, flower disc) using RT-PCR. RNA extraction, cDNA synthesis and primer design (Table S3) were performed as described above. PCR was performed in using 2 × GoldStar Best MasterMix (CWBIO, Beijing, China) according to its protocol. PCR products were visualized on a 1.2% agarose gel after electrophoresis.

Results

Transcriptome and Proteome analysis of jujube flowers

RNA sequencing of two independent flower samples (flower 1 and 2) on an Illumina Hiseq 2000 platform generated 52.8 and 59.4 million clean paired-end reads after quality control for flower 1 and 2, respectively (Table S1). In general, 90.1% of the reads from flower 1 and 92.2% of flower 2 were mapped onto the ‘Junzao’ jujube gene models, and 88.1% and 90.3% of reads were mapped to unique sites, respectively. A total of 18,646 (67.9%) and 17,995 (65.6%) genes had RPKM values ≥1 in two samples, respectively (Table S2). The two samples showed a strong correlation (Pearson r = 0.96, P < 0.01) on the RPKM metrics. In addition, the rank pattern of expression levels of the selected 30 genes generated by qRT-PCR analysis showed a linear correlation with their RPKM values (r = 0.95, P < 0.01). Taken together, both correlation analyses and independent qRT-PCR evaluation confirmed the reproducibility and validity of the relative expression levels based on the RPKM methods in the transcriptome analysis (Table S2).

Figure 1 Venn diagram of the number of expressed genes with RPKM ≥1 and the identified proteins in the proteome of jujube flowers.

In total, 80,802 MS/MS spectra were obtained and evaluated against the ‘Junzao’ jujube gene prediction database. We identified 7853 proteins with 42,925 unique peptides originating from 80,394 tandem mass spectrometry spectrum assignments with a false-discovery rate (FDR) below 1%, which accounts for nearly 30% of all predicted ‘Junzao’ gene models (27,443).

Comparison of the proteome and transcriptome

We compared the transcriptome and proteome data generated from the flower samples based on the ‘Junzao’ gene models. As shown in the Venn diagram (Fig. 1), 94% of genes identified by MS as protein could also be detected at the mRNA expression level (RPKM ≥1), and only 80 genes were missing. The identified proteins accounted for 41.2% of the transcripts with RPKM values ≥1.

The RPKM values of all detected genes show a bimodal distribution (Fig. 2A) in which 81.27% (17,995) and 64.08% (14,182) of genes had signals of RPKM values ≥1 and 5, respectively. When the criterion was set at 5, the left shoulder of the bimodal distribution line disappeared and transformed into a Gaussian distribution. By comparison, the identified protein abundance showed a broader distribution range than the filtered mRNA abundance (Figs. 2A and 2B). Among the 7,497 genes identified in the proteomic analysis, only 129 (1.6%) and 473 (6.0%) of genes had RPKM values <1 and 5, respectively. RPKM values of the genes identified in the proteome or those not identified differ clearly (Fig. 2C). We also found that the RPKM-based transcript abundance values correlated (Spearman’s correlation 0.5) well with iBAQ-based protein abundance values in jujube flowers (Fig. 2D). The top10 categories of GO for the proteome and the corresponding transcriptome in the three aspects of cell components, biological progress and molecular functions are shown in Fig. 3. The ranking patterns were generally similar in the transcriptome and proteome in the three aspects, and the highest-ranking categories were consistent among the three aspects.

Figure 2 Comparison of proteomics and RNA-Seq data generated from jujube flowers.

(A) Distribution of RPKM values in different levels. (B) Distribution of RPKM values of identified and unidentified genes in proteomic. (C) Distribution of proteins abundance (iBAQ intensities, FDR = 1%). (D) A density scatter plot of iBAQ versus RPKM values. The colour indicates the percentage points that were included in a region representing a specific colour.

Figure 3 Lists of ten highly ranked GO categories of proteome and their corresponding transcriptome.

(A) Biological progress, (B) cellular components, (C) molecular function.

Protein abundance

The iBAQ values are considered as estimates of the absolute amount of each protein. The most abundant 224 proteins (2.85%) constituted 50% of the jujube flower proteome, and the 691 most abundant proteins, representing 80% abundance, dominated the total protein abundance (Fig. 4). The 41 most abundant proteins comprised 20% of the proteome. The top7 proteins with iBAQ values >1010 included two non-specific lipid transfer proteins, histones, actin-related proteins, fructose-bisphosphate aldolase, Bet v I type allergens and unknown proteins. However, ribosomal proteins with six members showed the most abundant protein mass (50, 381, 000, 000) within the top20 proteins (Supplementary Data).

Abundance values of each protein reflect their contribution to the total proteome (Nagaraj et al., 2011). As shown in the GO category, more members of those proteins did not necessarily possess a higher proportion of the protein mass (Fig. S1A). Some protein classes contributed a significantly higher proportion of the total protein mass compared with the corresponding identified protein number. Based on the cellular components, we identified 175 (2.3%) different ribosomes accounting for 17% of the total protein mass in our data (Fig. S1A). Similarly, the intracellular components with 230 genes (3.1%) contributed 13.96% of the proteome mass. This large difference was also observed in the “structural constituents of ribosomes” belonging to ‘molecular function’ and ‘translation’ belonging to ‘biological progress’ (Figs. S1B and S1C). In contrast, integral membrane proteins accounted for 2.3% of the protein mass but contributed much less to the proteome (1.0% of total protein mass).

Figure 4 Cumulative protein mass from the highest to the lowest abundance proteins identified in jujube flowers.

Identification of SFB gene

We first identified seven candidate SFB genes (Table S3) in LG1 around S-RNase (Zj.jz035833030) by Blastp analysis. Phylogenetic analysis showed that S-locus F-box genes belonging to Plantaginaceae, Solanaceae, and Maleae clustered into one group. Jujube SFB genes were located between Prunus SFB genes and SLFL genes (Fig. 5). According to the transcriptomic analysis, we found that No. 5 (Zj.jz026591005) and No. 7 (Zj.jz039541036) were slightly expressed in budding flowers while the other four genes were expressed in several tissues (Fig. 6A). RT-PCR analyses showed that Zj.jz039541036 was strongly expressed in pollens while Zj.jz026591005 was only weakly detected in all flower organs. In contrast to RT-PCR results, Zj.jz027555002 was not detected in any of the tissues (Fig. 6B). Therefore, we consider Zj.jz039541036 as the candidate SFB gene of Z. jujuba.

Figure 5 Phylogenetic analysis of the S locus-linked F-box genes from different taxa exhibiting S-RNase-based GSI and seven F-box genes of Ziziphus jujuba located in the same chromosome with the S-RNase gene.

The neighbor-joining tree was constructed based on aligned amino acid sequences corresponding to those S locus-linked F-box genes from well characterized S haplotypes in each plant species using Mega 6.0 (bootstrap = 1,000). Zj. jz039541036 was the pollen-specificity candidate SFB gene of Jujube.

Figure 6 (A) Transcriptomic and (B) RT-PCR analysis of candidate S-locus F-box genes expression in different organs. fruit 1: young fruit, fruit 2: red fruit, flower: opening flower.

The number of each photo corresponds to the F-box genes listed in Table S2.

Discussion

In this study, we used an FASP-SCX-LC-MS/MS method to resolve the proteome of jujube flower organs. In total, we identified 7497 proteins that accounted for nearly 30.0% of all predicted ‘Junzao’ gene models (27,443). This depth was similar to the results of the shotgun proteome study of Arabidopsis flowers (Baerenfaller et al., 2008), in which 7040 proteins (26.1% of all gene models) were identified and represented nearly 50% of all identified Arabidopsis genes in the proteome map. Compared to the improved LC-MS/MS technologies, two-dimensional electrophoresis/mass spectrometry (2-DGE/MS) and multi-dimensional protein identification technology (MudPIT) approaches only identified 2,400 floral proteins in wild Arabidopsis flowers (Feng et al., 2009). Lack of genome resources could undermine the efficiency of protein identification. In one study, only 1,488 proteins were obtained from the fruit of Punica granatum by nanoliquid chromatography high-resolution tandem mass spectrometry (Capriotti et al., 2013), and 554 of the 2,043 peptides could not be annotated in the proteome of the midgut of Spodoptera litura larvae by shotgun ESI-MS (Liu et al., 2010). In the present study, 58% of the genes with RPKM values ≥1 were still not covered by the proteome in jujube flowers, suggesting the limitation of the present research strategies. The flower is a complex plant organ that secretes many sugars that likely interfere with protein separation (Righetti & Boschetti, 2016). When the flower is separated into different organs, such as pistil, petal, stigma, and style, the depth of the flower proteome may be further increased. In general, plant proteomics often identify fewer proteins than animal and human cell or organ proteomics because of higher amount of proteins in animals compared to the protein levels in plant biomass (for example, cell walls contain 90–95% polysaccharides and 5–10% proteins). Therefore, a new strategy for protein extraction should be considered (e.g., CPLL; Righetti & Boschetti, 2016).

Recent advances in genome sequencing and proteomics make it possible to study protein abundance and its relationship to corresponding mRNA levels. The distribution pattern of all transcripts in transcriptome showed a bimodal distribution. Transcripts in the left part of bimodal distributions (Fig. 2A) possessed less than one copy and were not expressed as functional protein in tissues (Hebenstreit et al., 2011). When the RPKM value criterion increased to 5, the RPKM values of transcripts showed a Gaussian distribution and were close to the distribution of genes detected in the proteome (Fig. 2A). In addition, the expression levels of those genes that have their corresponding proteins identified in the proteome were generally higher than those of which the proteins were not detected in the proteome (Fig. 2B). Genome-scale proteome analysis of A. thaliana preferentially detected proteins that are expressed at higher transcript frequencies (Baerenfaller et al., 2008). Feng et al. (2009) also found that proteins encoded by genes with relatively high levels of expression were detected with greater frequency in the proteome of wild Arabidopsis flower investigated by 2-DGE/MS and MudPIT approaches. These results confirm that genes with higher expression levels have a high likelihood of being translated into active protein.

Unlike the observed domination of RuBisCO (ribulose-1,5-bisphosphate carboxylase/oxygenase) in leaf proteins (>50%) and of storage proteins in seeds (>80%), in the jujube flower proteome, ribosomal proteins were the most abundant protein class, accounting for 17% of the total protein mass. This reflects the different biological functions of floral organs in the role of protein composition. Other abundant proteins also contribute to the special features of flowers. Non-specific lipid transfer proteins are abundant and widely distributed in plants, representing as much as 4% of total soluble proteins (Liu et al., 2015). Plant actin-related protein are also an important component and are widely distributed in plant cells. Fructose-bisphosphate aldolase is a key enzyme involved in glycolysis, gluconeogenesis in the cytoplasm, and in the Calvin cycle. Members of fructose-bisphosphate aldolase that are highly expressed in flowers may play an important role in pollen tube growth (Lu et al., 2012; Tang, 2013). In addition, abundant Bet v I type allergens make jujube flowers an allergen source for humans. This finding could also characterize the feature of jujube flower based on proteomics.

GSI is an important feature of reproductive biology in core-eudicots and exhibited in Z. jujuba. It has been well characterized that GSI is controlled by one pistil-specificity S-RNase and a single (or multiple) pollen–specificity F-box gene(s). We found there were some differences between the RNA-Seq and RT-PCR. According to the transcriptomic analysis, we were failed to identify a flower-specificity F-box gene linked to S-RNase based on the transcriptomic analysis (Fig. 6A). In contrast, we found a single pollen-specific F-box gene (Zj.jz 039541036) based on RT-PCR analysis after dissecting the flower into its component organs. Most likely this is because the RNA-Seq of whole flowers is not reflecting the gene expression level in specific organs or tissues. Phylogenetic analysis revealed that the jujube SFB gene was close to that of Prunus (Fig. 5), in which GSI exhibited a distinct system of recognition between pollen and pistil involving a single SFB gene. This contrasts with multiple F-box genes that are activated in other plants (Matsumoto & Tao, 2016). With the formerly identified S-RNase gene in Z. jujuba, therefore, we concluded that jujube GSI behaviour was like Prunus and controlled by the S-RNase and a single SFB gene together. This finding will facilitate further characterization of the underlying molecular mechanism of the reproductive biology of Z. jujuba.

Supplemental Information

Figure S1 Gene ontology analysis showing the proportion of identified proteins and the proportion of the protein mass that is attributed to these annotations

Click here for additional data file.

Table S1 Statistics of clean reads after quality control and the data mapped to genome of Junzao

Click here for additional data file.

Table S2 q-RT-PCR result of the selected genes

Click here for additional data file.

Table S3 Candidate SFB genes and the designed primer sets for RT-PCR analysis

Click here for additional data file.

Abbreviations

LC-MS/MS Liquid Chromatography-tandem Mass Spectrometry

MS Mass Spectrometry

RPKM Reads Per Kilobase per Million mapped reads

FDR False discovery rate

CBB Coomassie Brilliant Blue

SDS-PAGE Sodium Dodecyl Sulphate Polyacrylamide Gel Electrophoresis

MW Molecular Weight

GO Gene Ontology

qRT-PCR Quantitative real-time reverse transcription-PCR

iBAQ Intensity Based Absolute Quantification

Additional Information and Declarations

Competing Interests

Author Contributions

Data Availability

The authors declare there are no competing interests.

Ruihong Chen performed the experiments, analyzed the data, contributed reagents/materials/analysis tools, wrote the paper, prepared figures and/or tables, reviewed drafts of the paper.

Guoliang Chen performed the experiments, reviewed drafts of the paper.

Jian Huang conceived and designed the experiments, performed the experiments, analyzed the data, contributed reagents/materials/analysis tools, wrote the paper, prepared figures and/or tables, reviewed drafts of the paper.

The following information was supplied regarding data availability:

The raw data has been supplied as a Supplementary File. and the raw-sequence reads data were deposited in NCBI Sequence Read Archive (SRX1518648 and SRX1518646).

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
