# Peer review of "Shot-gun proteome and transcriptome mapping of the jujube floral organ and identification of a pollen-specific S-locus F-box gene"

_PeerJ, doi:10.7717/peerj.3588_

## Round 0.1 · original submission · Major Revisions

I agree with the comments of two reviewers mostly. I want to emphasize the following points:

(1) The authors need to bring better linking of their results to flower biology and make the biological story supported by the data.

(2) one replicate for RNA-Seq and Proteomics is definitely a limitation for this study. In the discussion part, the authors need to address the relevant concern.

(3) The manuscript needs thorough professional English editing.

Reviewer 1 ·

Basic reporting

Additional proofreading from native English speaker is recommended to avoid some of the language problems.

Experimental design

The selection of analytical methods is reasonable, sufficient technical details have been reported. However, the evaluation or prior knowledge regarding to the reproducibility of the techniques should be described in the manuscript

Validity of the findings

Not many statistical components involved as the authors only have a couple biological replicates.

Additional comments

The manuscript titled ‘Shot-gun proteome and transcriptome mapping of jujube floral organ’ conducted a survey of protein and RNA profiles of Jujube floral organ using high throughput ‘omics’ techniques, and generated huge amount of data. The study is carefully designed, the techniques used are reasonable, and the data interpretation is mostly clear. I recommended the following revisions to be made before the manuscript can be accepted for publication.
Major comments:
1. It is not clear how reproducible or robust the applied techniques can achieve. The authors only collected two biological replicates from the plant, and one was used for protein analysis and the other one was used for transcriptomics analysis. Therefore, a statement either based on their prior knowledge or the existing literature about the reproducibility of their work should be added.
2. The author mentioned that they collected samples from different trees, and Table S1 only displayed the comparison of two samples, with roughly 12% difference in total number of reads. How’s the data from other trees comparing to these reported two samples, would this variation of 12% a general expectation of their biological reproducibility?

Minor comments:
• Need to give full name for ‘RPKM’ in the abstract.
• The English language should be improved to ensure that your international audience can clearly understand your text.
• line 25 and line 26. Past tenses for “reflect” and “provide” are required.
• Line 57 “The increasing” should be “Increasing” and line 58 “The flower is” should be “Flowers are”. Use of “a”, “an”, and ”the” should be careful.
• Some typos should be avoided. For example in Line 277 ”30%in” , a space is missing.
• Figure 1, the caption is hard to understand. What does “expressed genes” mean? Protein coding or all known genes in a “Junzao” model
• Figure 4 is not clear enough. Are these number of proteins exactly corresponding to 20%, 40% and 60% proteins?

Reviewer 2 ·

Basic reporting

* Numerous typo's in the text:
eg: line162, twice the word 'for' ; line 217&233&259&277&284: missing spaces
* line 176: "to count the read numbers mapped" should read: "to count the reads mapping to"
* RPKM is reads per kilobase per millions reads mapped (line 177 & 34)
* line 215: I'm not sure but I guess the authors mean "two INdependent samples"?
* line 277: should read: "...which accounts for nearly 30% of all predicted ..."
* not sure what happened with fig1 but the number in one of the sections in hardly readable
* there is a space too much in the legend of fig3 "C ellular"
* I don't see any indication where the authors deposited their raw data (SRA? ...)
* The are some discrepancies in the text eg. in the introduction the authors mention they sequenced using a HiSeq2000, somewhat further in the text they talk about HiSeq2500 ?
* line 347: Use verbs in past sense

Experimental design

* The authors fail to tell anything on the biology of flowers in jujube. Most of the text is highly descriptive, they should try to do a more in depth analysis of the obtained result and better link it to the flower biology
* In the methods it’s not described what kind of sequencing has been performed, read length? Paired/single end?
* Nowhere it is mentioned how or where the authors got their GO assignments for the jujube proteins?

Validity of the findings

* on line 218, the authors mention that x% of reads mapped to other unique sites? What do they mean by that?
* lines 305-307: it is stated that the expression of identified genes in the proteome is higher than the unidentified ones. This is somewhat confusing to me, what do the authors mean with ‘unidentified in the jujube flowers’?

---

## Round 0.2 · Major Revisions

The author needs to seriously address comments from Review #2 and your responses should not be limited to solely the examples identified by the reviewers.

Again, professional English editing is also critical for me to accept the manuscript eventually for publication.

Reviewer 1 ·

Basic reporting

The authors have addressed my questions and concerns.

Experimental design

No comment.

Validity of the findings

No comment.

Reviewer 2 ·

Basic reporting

I must admit that the manuscript improved compared to the previous version. However despite the fact the authors did acknowledge most of the reviewers comments, several issues with the language applied are still present.
eg. missing spaces: line 257, line 307 ....
miss use of the word 'respectively'
these and other issues are especially present in the new parts added to the original manuscript: line 302: the word "posited" ?? line 305: ...that Zj.jz... were" should be "...was"

Experimental design

I'm still wondering whether the use of the RPKM threshold of 1 is valid? Yes, it is valid as such but I seem to think that usually people choose a higher threshold, eg. 10?

Validity of the findings

no comment

Additional comments

While the authors did act on most of my comments on the previous version, some of them have not been reacted on , eg. where did the authors got their GO data from? The Venn diagram figure is still unreadable. Caption of fig 3 still it reads " C ellular" . Also the newly added paragraphs should be thoroughly checked for english spelling and grammar.
Line 239: it mentions 'single end reads" while in the methods section it says paired-end?
line 232: I assume the authors mean qRT-PCR (and not RT-PCR)?
line 221: "we used those known S-locus" ... which "those"??
line 228: missing "on" in "... calculated based the ..."

---

## Round 0.3 · Minor Revisions

Please improve the English language of the manuscript. Reviewer 2 has put good efforts in editing your manuscript (see Annotated manuscript).

Reviewer 2 ·

Basic reporting

language used is still an issue

Experimental design

no comments

Validity of the findings

no comments

Additional comments

I fear that unfortunately (and apparently despite the efforts being put into it) the text is language-wise not publication ready.
In stead of keep on listing the issues I observe, I changed the document myself using track-changes. Please go trough it and accept or reject the changes as you wish.

---

## Round 0.4 · accepted · Accept

The paper is ready for publication.